# A Heterostructure Photoelectrode Based on Two-Dimensional Covalent Organic Framework Film Decorated TiO_2_ Nanotube Arrays for Enhanced Photoelectrochemical Hydrogen Generation

**DOI:** 10.3390/molecules28020822

**Published:** 2023-01-13

**Authors:** Yue Zhang, Yujie Li, Jing Yu, Bing Sun, Hong Shang

**Affiliations:** School of Science, China University of Geosciences (Beijing), Beijing 100083, China

**Keywords:** two-dimensional materials, covalent organic framework, surface modification, heterostructure, photocatalytic hydrogen generation

## Abstract

The well-defined heterostructure of the photocathode is desirable for photoelectrochemically producing hydrogen from aqueous solutions. Herein, enhanced heterostructures were fabricated based on typical stable covalent organic framework (TpPa-1) films and TiO_2_ nanotube arrays (NTAs) as a proof-of-concept model to tune the photoelectrochemical (PEC) hydrogen generation by tailoring the photoelectrode microstructure and interfacial charge transport. Ultrathin TpPa-1 films were uniformly grown on the surface of TiO_2_ NTAs via a solvothermal condensation of building blocks by tuning the monomer concentration. The Pt_1_@TpPa-1/TiO_2_-NTAs photoelectrode with single-atom Pt_1_ as a co-catalyst demonstrated improved visible-light response, enhanced photoconductance, lower onset potential, and decreased Tafel slope value for hydrogen evolution. The hydrogen evolution rate of the Pt_1_@TpPa-1/TiO_2_-NTAs photoelectrode was five times that of Pt_1_@TpPa-1 under AM 1.5 simulated sunlight irradiation and the bias voltage of 0 V. A lower overpotential was recorded as 77 mV@10 mA cm^−2^ and a higher photocurrent density as 1.63 mA cm^−2^. The hydrogen evolution performance of Pt_1_@TpPa-1/TiO_2_-NTAs photoelectrodes may benefit from the well-matched band structures, effective charge separation, lower interfacial resistance, abundant interfacial microstructural sites, and surficial hydrophilicity. This work may raise a promising way to design an efficient PEC system for hydrogen evolution by tuning well-defined heterojunctions and interfacial microstructures.

## 1. Introduction

Hydrogen is regarded as a renewable, eco-friendly, and high-density energy source and has been attractive in the fields of both science and engineering to address the crises of energy shortage and environmental pollution [1,2]. Benefiting from the inexhaustible solar energy and plentiful water, solar-driven hydrogen evolution from water is regarded as a potential candidate to generate storable fuels for the future sustainable energy strategy without the carbon footprint [3]. A designable photocatalyst is the key component for converting solar energy to produce hydrogen with desirable efficiency [4,5,6,7]. Recently, two-dimensional covalent organic frameworks (2D COFs), as well as metal-organic frameworks (MOFs), have emerged as promising materials in hydrogen evolution due to their predictable structures, tunable π-conjugation, high crystallinity, and permanent pore channels [8,9,10,11,12]. The precisely tailored 2D COFs provide a platform to illustrate the structure–property relationships during the hydrogen evolution reaction. Various strategies have been developed to enhance the light-harvesting rate and charge separation efficiencies, such as band-gap engineering [13,14], donor–acceptor system fabrication [15,16,17,18], and linkage construction [19]. Their high crystallinity further facilitates rapid charge diffusion, ensuring optimal charge percolation. The abundant heteroatoms and high specific surface area of the frameworks provide substantial active interfaces for loading suitable co-catalysts and the reactant accessibility for catalyzing redox reactions [20]. The tailored coordinating microenvironments of 2D COFs benefit the immobilization of single-atom noble metals to reduce the overpotential, promoting the hydrogen evolution procedure [21]. Furthermore, the pre-designed heterostructures and nanocomposites were explored with further-improved separation efficiency of the photogenerated charge carriers by combining 2D COFs with other motifs such as inorganic semiconductors [22,23,24,25,26], metal-organic frameworks [27], 2D materials [28,29], etc. However, most of the 2D COF-based photocatalysts were suspended in reacting systems for hydrogen evolution, suffering from challenges of poor powder dispersibility, recycling inconvenience, and discontinuous production of hydrogen.

Photoelectrochemical (PEC) catalysis provides another cost-effective and promising route for converting water into sustainable hydrogen under solar light irradiation, which is apt to overcome the aforementioned drawbacks of photocatalysts [30]. Integrating 2D COF-based photocatalysts into PEC cells was primarily performed by coating a suspended COF slurry in organic solvents onto substrate electrodes [31,32,33]. On the other hand, oriented films of conjugated 2D COFs grown on transparent conducting fluorine-doped tin oxide and indium tin oxides were also reported as photocathodes for generating hydrogen [34]. Recently, solution-processed COF nanoplates were proposed to construct centimeter-scale-homogeneous COF films for fabricating a complex cascading photoelectrode for PEC hydrogen evolution [35]. Although these efforts have been made, constructing the 2D COF-based PEC system for hydrogen generation remains in the early stages of development.

Inspired by the advanced features of 2D COF-based photocatalysts and the desirable architecture of the PEC system for hydrogen evolution, we proposed the in situ fabrication of single-atom Pt (Pt_1_) loaded TpPa-1 films on TiO_2_ nanotube arrays (NTAs)/Ti foil (Pt_1_@TpPa-1/TiO_2_-NTAs photoelectrode, as shown in Figure 1) for PEC hydrogen evolution. Here, the cost-effectively anodized TiO_2_ NTAs were employed as substrates for growing COF films due to the stable and fantastic PEC performance for solar-driven water splitting [22,23]. The stable and ultrathin TpPa-1 COF films were chemically fabricated on the anodized TiO_2_ NTAs to prove the concept of conveniently integrating effective charge transfer heterostructure and interfacial catalytic sites in a single photoelectrode for PEC hydrogen evolution. By integrating COF-based heterostructures into the photoelectrode, the photoelectrode could inherit the cascading charge separation from COF-based heterostructures and address the challenges of poor powder dispersibility, recycling inconvenience, and discontinuous production of hydrogen in the application of COF-based photocatalysts. The surficial microstructures of TiO_2_ NTAs and the thickness of TpPa-1 films were also discussed as ways of impacting the PEC hydrogen evolution performance. The spectral investigation indicated the extended absorption, enhanced photocurrent density, and reduced interfacial resistance of TpPa-1/TiO_2_-NTAs heterostructures. The Pt_1_@TpPa-1/TiO_2_-NTAs photoelectrode demonstrated lower overpotential, higher current density, and enhanced PEC hydrogen generating performance. This work may shed light on the design and fabrication of heterostructure-based photoelectrode for PEC hydrogen evolution. The convenient fabrication of Pt_1_@TpPa-1/TiO_2_-NTAs photoelectrode would also provide a facile and potential way of improving their industrial applications.

## 2. Results and Discussion

### 2.1. Fabrication and Characterization of Pt_1_@TpPa-1/TiO_2_-NTAs Photoelectrode

The Pt_1_@TpPa-1/TiO_2_-NTAs photoelectrode is fabricated starting from the synthesis of TiO_2_ NTAs via the anodization-annealing method. The morphology and crystalline phase of the obtained TiO_2_-NTAs are confirmed by SEM observation and XRD analysis. The typical discrete nanotubes are obtained with an average diameter of 40.55 ± 8.46 nm after the anodization time of 0.25 h (Figure 2a). Few nanoparticles (40–50 nm) distribute on the surface of TiO_2_ NTAs. More nanoparticles emerge when the anodization time is 0.5 h, and tend to fuse covering on the top of nanotubes with the enlarged diameter (~69.60 nm, Appendix A). The surficial nanoparticles grow larger at a longer anodization time forming a flat film (1 h or 1.5 h, Appendix A). The underneath nanotubes collapse as shown in the edges. Only nanoparticle aggregates are found on SEM images as the anodization time is 2 h. According to the XRD patterns (Appendix A), these TiO_2_ nanostructures reveal a typical anatase phase matching well with that of JCPDS No. 21-1272 coupling with some diffraction peaks of Ti foil substrate (JCPDS No. 44-1294). The diffraction intensity of the TiO_2_ NTAs is weak at the anodization time of 0.25 h indicating the thinner oxidization layer with lower crystallinity and plenty of defects, and then gradually increases as prolonging the anodization times. UV–vis spectra demonstrate the common absorption of formed TiO_2_ nanostructures to ultraviolet light (Appendix A), while the absorption band and edge show a blue shift indicating the increased band gap as prolonging the anodization times. The highest photocurrent response and lowest impedance resistance are also obtained at the anodization time of 0.25 h (Appendix A). The longer the anodization time, the lower the photocurrent density and larger impedance resistance. The outstanding PEC performance of TiO_2_ NTAs at the anodization time of 0.25 h may be attributed to the smaller band-gap, higher absorbance to light irradiation, regular nanotube morphology for the mass delivery and thin oxidized TiO_2_ layers for charge transfer with Ti substrate electrode. Considering all the above results, the TiO_2_ NTAs obtained at the anodization time of 0.25 h are selected for further investigation.

TpPa-1 COF films are constructed on the surface of amino-modified TiO_2_ NTAs via the solvothermal polymerization of monomers with various concentrations. The surface of TpPa-1/TiO_2_-NTAs obtained with 0.2 mM Tp and 0.3 mM Pa reveals similar morphology and nanotube size compared to that of TiO_2_ NTAs (Figure 2b). This indicates that the TpPa-1 COF films tightly grow along the top and inner surface of TiO_2_ NTAs, forming extensively-spreading TpPa-1/TiO_2_-NTAs heterostructures. The TiO_2_ NTAs can be still clearly observed when the concentration of Tp is increased to 1.5 mM (2.25 mM for Pa). The open nanotube would facilitate the mass transfer combined with the high surface area and porous structures of TpPa-1 films. However, the nanotube diameter gradually shrinks (Appendix A). The TpPa-1 films grow continuously over the whole TiO_2_ NTAs surface, filling the nanotubes of TiO_2_ structures and further increasing the monomer concentrations. No typical diffraction peaks of TpPa-1 films are detected in the XRD patterns of TpPa-1/TiO_2_-NTAs (Appendix A) attributed to the ultrathin layer on TiO_2_ NTAs with the lower monomer concentration.

The chemical structures of TpPa-1 films are verified by FTIR and XPS spectra. As shown in Figure 2e, the stretching modes of N−H bonds in Pa (3100~3400 cm^−1^) and aldehyde C=O stretching bands of Tp at 1635 cm^−1^ disappear in the spectra of both TpPa-1 powders and TpPa-1/TiO_2_-NTAs. This result indicates the total consumption of monomers during the formation of TpPa-1 films. No characteristic hydroxyl (O−H) and imine (C=N) stretching vibration modes are recorded in TpPa-1 FTIR spectra, while a broadening shoulder band at 1610 cm^−1^ emerges assigned to the keto C=O in TpPa-1 films due to the extended conjugation and strong intramolecular hydrogen bonds [36]. A strong peak around 1582 cm^−1^ arises from the C=C stretching vibration mode of the formed keto configuration. TpPa-1 films synthesized from higher monomer concentrations demonstrate similar FTIR spectra with the same stretching modes (Appendix A). These results reveal the formation of TpPa-1 films on TiO_2_ NTAs, which is also confirmed by the element distribution of SEM (Figure 2d). Remarkable carbon and nitrogen elements locate around the TiO_2_ NTAs, profiling the nanotube apertures compared to the homogeneously distributed Ti and oxygen elements. The formation of TpPa-1 films is also verified by the XPS survey spectra (Figure 2f). The assignments of C1s, N1s, and O1s can well match the chemical structure of TpPa-1 COF (Figure 2g,h and Appendix A). The weak C1s and N1s peaks compared to Ti2p and O1s indicate the thinner TpPa-1 films formed on TiO_2_ NTAs.

The Pt_1_@TpPa-1/TiO_2_-NTAs photoelectrode is finally constructed by photo-depositing Pt species to a TpPa-1/TiO_2_-NTAs photoelectrode. The photo-deposited Pt does not change the morphology of TpPa-1/TiO_2_-NTAs (Figure 2c) and uniformly distributes on TpPa-1/TiO_2_-NTAs based on the Pt element mapping (Figure 2d). The typical Pt4f and Pt4d bands are indexed from the survey spectrum of Pt_1_@TpPa-1/TiO_2_-NTAs (Figure 2f). The high-resolution spectra of Pt4f cores are assigned to two bands at 74.91 eV for Pt(IV)4f_7/2_ and 78.25 eV for Pt(IV)4f_5/2_ (Figure 2i). The other two peaks are located at 72.83 eV and 76.24 eV attributed to the Pt4f_7/2_ and Pt4f_5/2_ of single-atom Pt (denoted as Pt_1_), respectively, according to the previous work [21]. No Pt clusters or nanoparticles are formed during the photo-deposition based on the above results. The N1s core can be assigned to the formed C−N band at 399.18 eV and protonated N at 401.37 eV, which slightly shifts to a higher binding energy due to the coordination of Pt_1_ with N and O moieties (Figure 2h). All the above results indicate the successful construction of the Pt_1_@TpPa-1/TiO_2_-NTAs photoelectrode.

The interfacial hydrophilicity of the Pt_1_@TpPa-1/TiO_2_-NTAs photoelectrode is an important factor in effectively producing hydrogen from water. The contact angle slightly increases to 35° in the case of TpPa-1/TiO_2_-NTAs from 27° of TiO_2_ NTAs and recovers to 26° of Pt_1_@TpPa-1/TiO_2_-NTAs (Appendix A). The hydrophilicity of Pt_1_@TpPa-1/TiO_2_-NTAs photoelectrodes may be attributed to the polarized keto form and protonated linkages of thinner TpPa-1 films, benefiting the higher affinity to water and protons leading to excellent hydrogen evolution efficiency [37,38]. Additionally, the TpPa-1/TiO_2_-NTAs photoelectrode demonstrates higher thermal stability based on the thermogravimetric analysis (Appendix A). These characteristics would promise the potential performance of the Pt_1_@TpPa-1/TiO_2_-NTAs photoelectrode.

### 2.2. Photoelectrochemical Performance of Pt_1_@TpPa-1/TiO_2_-NTAs Photoelectrodes

The absorption of TpPa-1/TiO_2_-NTAs photoelectrodes to light irradiation is evaluated by the UV–vis absorption spectra (Figure 3a). The TpPa-1/TiO_2_-NTAs show a broad absorption band covering ultraviolet and most visible light spectrum with an absorption edge of 580 nm, which is inherited from the excellent absorption nature of TpPa-1. The optical band gaps of TiO_2_ NTAs and TpPa-1 are evaluated as 3.17 eV and 2.02 eV, respectively, via the Kubelka-Munk equation based on the absorption spectra (Figure 3b). When the monomer concentrations are increased, the obtained TpPa-1/TiO_2_-NTAs demonstrate similar absorption characteristics (Appendix A).

The PEC performance of the Pt_1_@TpPa-1/TiO_2_-NTAs photoelectrode is further evaluated. As shown in Figure 3c, TpPa-1/TiO_2_-NTAs (0.2 mM for Tp) demonstrate enhanced photocurrent density (1.14 mA cm^−2^) compared to TiO_2_-NTAs (0.92 mA cm^−2^) under the full-range light irradiation, implying the improved photo-induced charge separation between the TpPa-1/TiO_2_-NTAs heterostructures. The effective charge separation may also be attributed to the enhanced photoconductance of TpPa-1/TiO_2_-NTAs as revealed by EIS spectra (Figure 3d and Appendix A). The resistance of TpPa-1/TiO_2_-NTAs is estimated as below 1 kΩ, which is smaller than that of TiO_2_ NTAs (>1.8 kΩ). However, the TpPa-1/TiO_2_-NTAs synthesized at the higher monomer concentrations reveal the remarkably decreased photocurrent density (Appendix A), which may originate from the increased interfacial resistance for charge transfer due to the inherent low conductivity of TpPa-1 (Appendix A). Loading Pt_1_ to TpPa-1/TiO_2_-NTAs further leads to enhanced photocurrent density (1.63 mA cm^−2^) and lower interfacial resistance (242 Ω). The photocurrent density is higher than the reported counterpart in previous works.[27,28,29,30] Furthermore, a steady photocurrent density is recorded after two light-on/off rounds for 3 h (Appendix A), indicating the highly stable PEC response of the Pt_1_@TpPa-1/TiO_2_-NTAs photoelectrode. The excellent PEC performance of the Pt_1_@TpPa-1/TiO_2_-NTAs photoelectrode would benefit its application for hydrogen evolution.

### 2.3. Photoelectrochemical Hydrogen Evolution Based on Pt_1_@TpPa-1/TiO_2_-NTAs Photoelectrode

The PEC hydrogen evolution performance of the Pt_1_@TpPa-1/TiO_2_-NTAs photoelectrode is investigated in a PEC cell under light irradiation. LSV measurements reveal that the TpPa-1/TiO_2_-NTAs photoelectrode exhibits lower potential for proton reduction compared to TiO_2_-NTAs under light irradiation; this is attributed to the effective charge separation in the TpPa-1/TiO_2_-NTAs heterostructures (Figure 4a). With the higher monomer concentrations, the TpPa-1/TiO_2_-NTAs photoelectrodes demonstrate the increased reduction potential and decayed current density (Appendix A). A thinner TpPa-1 film seems desirable to fabricate on TiO_2_ NTAs for enhanced PEC performance by descending the interfacial resistance. On the other hand, Pt_1_@TpPa-1/TiO_2_-NTAs show a lower reduction potential and higher current density, indicating the enhanced interfacial catalytic property of photoelectrodes with Pt_1_ loaded. The overpotentials of Pt_1_@TpPa-1/TiO_2_-NTAs photoelectrodes are evaluated as 77 mV@10 mA cm^−2^ and 180 mV@50 mA cm^−2^, which are significantly lower than that of TpPa-1/TiO_2_-NTAs and TiO_2_-NTAs (Figure 4b). The brilliant PEC catalytic performance of Pt_1_@TpPa-1/TiO_2_-NTAs photoelectrode for proton reduction is also confirmed by its smaller slope value via Tafel plotting (Figure 4c). Therefore, the extended light absorption, lower charge transfer resistance, robust interfacial catalytic ability, and hydrophilic surface of Pt_1_@TpPa-1/TiO_2_-NTAs photoelectrode promise its excellent PEC hydrogen evolution performance in aqueous solutions.

The hydrogen evolution capacity of the Pt_1_@TpPa-1/TiO_2_-NTAs photoelectrode is evaluated by quantitative analysis. As expected, the Pt_1_@TpPa-1/TiO_2_-NTAs photoelectrode shows a remarkably enhanced hydrogen-generating capacity under light irradiation compared to Pt_1_@TpPa-1 and Pt@TiO_2_-NTAs (Figure 4d). Almost no hydrogen generates in the case of Pt@TiO_2_-NTAs. The Pt_1_@TpPa-1/TiO_2_-NTAs obtained from 0.2 mM Tp and 0.3 mM Pa show superior capacity for hydrogen generation compared to other higher monomer concentrations (Figure 4f). These results are accordant with the PEC performance of Pt_1_@TpPa-1/TiO_2_-NTAs and TpPa-1/TiO_2_-NTAs photoelectrodes. The hydrogen evolution rate of Pt_1_@TpPa-1/TiO_2_-NTAs ascends when the initial amount of K_2_PtCl_6_ alters from 0.5 mg to 1 mg and keeps steady as Pt dosage increases further (Figure 4g). A dosage of 1 mg of K_2_PtCl_6_ (1 mL, 1 mg mL^−1^) is used for photo-depositing Pt_1_ onto TpPa-1/TiO_2_-NTAs. The effect of applied bias voltages on the hydrogen evolution of Pt_1_@TpPa-1/TiO_2_-NTAs is also investigated in the PEC cell setup. The positive bias voltage, applied via an electrochemical station, can slightly promote and stabilize the hydrogen evolution rate of Pt_1_@TpPa-1/TiO_2_-NTAs photoelectrodes, while an attenuate capacity is recorded under the inversed voltage (Figure 4h), which agrees with the transient photocurrent response (Appendix A). These results indicate that Pt_1_@TpPa-1/TiO_2_-NTAs photoelectrode can effectively produce hydrogen from aqueous solutions even at 0 V and a suitable bias voltage. Under the optimized conditions, the Pt_1_@TpPa-1/TiO_2_-NTAs photoelectrode can continuously generate hydrogen from aqueous solutions without a significantly-descended hydrogen evolution rate for four cycles. The hydrogen evolution rate of Pt_1_@TpPa-1/TiO_2_-NTAs is evaluated as 0.5 nmol h^−1^ cm^−2^, about 5 times that of Pt_1_@TpPa-1 (Figure 4e). The photoelectrode can be conveniently reused by just being activated in THF for 4 h, averting the collection of photocatalysts from reaction solutions. The steady hydrogen generating rate of 6 h for 4 cycles, coupled with the stable photocurrent density response, also implies the long lifetime of Pt_1_@TpPa-1/TiO_2_-NTAs photoelectrode. All these results indicate the stable and robust properties of Pt_1_@TpPa-1/TiO_2_-NTAs photoelectrode for PEC hydrogen evolution.

### 2.4. Photoelectrochemical Hydrogen Evolution Mechanism

To further understand the PEC hydrogen evolution process, the charge transfer in the TpPa-1/TiO_2_-NTAs heterostructures and interfacial reaction is illustrated combined with the PEC performance of Pt_1_@TpPa-1/TiO_2_-NTAs photoelectrode. The flat-band potentials of TiO_2_ NTAs and TpPa-1 are evaluated as −0.46 V and −0.92 V (vs. SCE), respectively, based on the Mott–Schottky curves (Appendix A). The band diagrams of TpPa-1/TiO_2_-NTAs heterostructures are demonstrated in Figure 5a, with their optical band gaps. The lowest unoccupied molecular orbital (LUMO) of TpPa-1 and conduct band of TiO_2_ NTAs are evaluated as −0.58 eV and −0.12 eV, respectively, which are available to generate hydrogen from water. Both TpPa-1 and TiO_2_ NTAs can be excited under full-range light irradiation. These band structures benefit the effective charge separation between TpPa-1/TiO_2_-NTAs heterostructures. The excited electrons in TpPa-1 can further transfer to TiO_2_ NTAs. The separated electrons further reduce protons to produce hydrogen with the Pt_1_ co-catalysts. Because of the difficulty in oxidizing water, a sacrificial electron donor (SA is used here) is usually added to PEC hydrogen production systems to regenerate the photosensitizer. The photogenerated holes in the TpPa-1 films and TiO_2_-NTAs can be captured by the sacrificial SA and electrons from the Ti substrate electrode. Ultrathin TpPa-1 films with regular pore size on TiO_2_-NTAs and the hydrophilicity of Pt_1_@TpPa-1/TiO_2_-NTAs photoelectrode are conducive to the penetration of protons, water, and ions to the reactive sites of the heterostructure surface. The highly specific surface area and lower crystallinity of TiO_2_-NTAs provide more anchoring surface vacancy sites for the adequate loading of single metal atoms with the further assistance of TpPa-1 coordinating sites, contributing to the higher atom utilization efficiency and hydrogen evolution on the surface of the Pt_1_@TpPa-1/TiO_2_-NTAs photoelectrode. Based on the hydrogen evolution performance of Pt_1_@TpPa-1/TiO_2_-NTAs photoelectrodes, the concept of integrating heterostructures into a single photoelectrode for PEC hydrogen evolution is verified. The synergistic effect of enhanced light absorption ability, improved charge separation efficiency, ultrathin mass transfer channels, abundant single-atom co-catalyst load, and good hydrophilicity facilitates the good hydrogen evolution performance of the Pt_1_@TpPa-1/TiO_2_-NTAs photoelectrode. The hydrogen evolution rate could be further enhanced by enlarging the photoelectrode area for promising industrial applications.

## 3. Materials and Methods

### 3.1. Fabrication of Pt_1_@TpPa-1/TiO_2_-NTAs Photoelectrode

The TiO_2_ NTAs were prepared by using a facile anodization method. All chemicals were used as received without any further purification. Typically, a Ti foil (Alfa Aesar, 99.8%, metal basis) with a thickness of 1 mm (10 mm × 35 mm) was cleaned by using the ultrasonic method in acetone, ethanol, and deionized water for 15 min in turn, and dried under nitrogen flow. The electrochemical anode oxidation was performed by employing Ti foil as the anode and a Pt foil (10 mm × 10 mm) as the counter electrode in a fresh ethylene glycol solution containing 0.1 wt% of NH_4_F (Sinopharm Chemical Reagent Co. Ltd., >96%) and 1 wt% H_2_O. A DC powder supplied the anodization at 60 V at various times. Then, the anodized Ti foil was annealed in a muffle furnace at 450 °C for 2 h to obtain the crystalline TiO_2_ NTAs on the Ti foil. The surface of the obtained TiO_2_ NTAs was further modified by dipping TiO_2_-NTAs/Ti foil in 15 mL toluene with 430 μL 3-aminopropyltrimethoxysilane (APTMS, Sigma-Aldrich, 97%) and refluxing at 80 °C for 6 h. The amino-modified TiO_2_-NTAs/Ti foil was rinsed with toluene and dried at 70 °C overnight.

The TpPa-1 films were constructed on the surface of TiO_2_ NTAs by using a solvothermal method, as shown in Figure 1. A precursor solution was prepared by suspending 1,3,5-triformylphloroglucinol (Tp, Innochem, 97%, 0.2 mM~2 mM) and *p*-phenylenediamine (Pa, Innochem, 97%, 0.3 mM~3 mM) in a vial containing 3 mL mesitylene, 3 mL dioxane, and 1 mL acetic acid solution (3 M, Acros Organics, >98%). The precursor solution was homogeneously mixed by ultrasonic treatment for 15 min and then transferred into a 25 mL Teflon autoclave container. A homemade scaffold was used to horizontally hold the TiO_2_-NTAs/Ti plate in the reacting solution, facing the bottom. The autoclave was sealed and heated at 120 °C for 72 h, growing TpPa-1 films on the TiO_2_ NTAs surface. The obtained TpPa-1/TiO_2_-NTAs/Ti plate was rinsed with tetrahydrofuran (THF), activated in THF for 24 h, and finally dried at 70 °C overnight under vacuum conditions. Pt loading was performed by dipping the TpPa-1/TiO_2_-NTAs/Ti plate in an H_2_PtCl_6_ (Sigma-Aldrich, ≥99.9%) aqueous solution (1 mg mL^−1^, 0.5 mL, 1 mL, 2 mL, and 4 mL) for 30 min under the irradiation of an Xe lamp (AM 1.5 simulated sunlight, 300 W, 100 mW cm^−2^) according to the previously reported in situ photo-deposition method [20]. The Pt_1_@TpPa-1/TiO_2_-NTAs/Ti plate was washed with deionized water and dried at 70 °C for further characterization and measurements.

### 3.2. Characterizations

The chemical structure of the as-prepared photoelectrode was verified by Fourier transform infrared (FTIR) spectra and X-ray photoelectron spectra (XPS). FTIR spectra were recorded in the range of 400~4000 cm^−1^ with an interval of 4 cm^−1^ on a PerkinElmer Frontier spectrophotometer in the attenuated total refraction (ATR) mode with an additional variable angle reflectance accessory under ambient conditions. XPS spectra were performed on a Thermo Fisher Scientific ESCALAB 250Xi spectroscope with Al Kα radiation (photon energy 1253.6 eV) as the exciting source at a working voltage of 12.5 kV. The crystalline features of the samples were characterized by using a Bruker D8 Advanced X-ray diffractometer with Cu Kα radiation (λ = 1.5416 Å), operated at 40 kV and 40 mA ranging from 1.5 to 80° with a speed of 2° min^−1^. The microstructures and element distribution of samples were observed on a JEOL JSM-7900F scanning electron microscope (SEM).

### 3.3. Photoelectrochemical Performance and Hydrogen Evolution

The absorption performance of the Pt_1_@TpPa-1/TiO_2_-NTAs photoelectrode was recorded on a PerkinElmer Lambda 750 UV–vis scanning spectrophotometer in diffuse reflection mode with an integrating-sphere accessary in the range of 200 to 800 nm. The photoelectrochemical measurements were carried out on a CHI-760E electrochemical station with or without light irradiation from the Xe lamp. A standard three-electrode system was used for measurement, where the Pt_1_@TpPa-1/TiO_2_-NTAs photoelectrode (electrode surface of 1 cm^2^) was employed as the working electrode, saturated calomel electrode (SCE) as the reference electrode, Pt foil (length of 10 mm, diameter of 1 mm) as the auxiliary electrode. The photocurrent response of photoelectrodes was performed in a phosphorus buffer solution (PBS, pH = 7) containing 10 mM sodium ascorbate (Sinopharm Chemical Reagent Co. Ltd., >96%). Linear sweep voltammetry measurements were performed from 0 V to −1.2 V (vs. SCE), with a scan rate of 5 mV s^−1^ in a 0.5 M H_2_SO_4_ solution. Mott–Schottky curves were recorded with the voltage range of ±1.5 V under the frequency of 1 kHz in 0.5 M Na_2_SO_4_ aqueous solution. Electrochemical impedance spectra (EIS) were carried out in a frequency range from 0.01 Hz to 100 kHz, with an amplitude of 5 mV in 0.5 M Na_2_SO_4_ aqueous solution.

The PEC hydrogen production was performed in a sealed PEC reacting cell made of stainless steel with a quartz window equipped with the standard three-electrode system, circulating water system, and PBS solution (50 mL, pH = 7, containing 0.01 M sodium ascorbate (SA)). The area of the working electrode (Pt_1_@TpPa-1/TiO_2_-NTAs) was fixed at 1 cm^2^. The electrolyte solution was degassed for 30 min with high-purity nitrogen floating under dark conditions. Hydrogen was generated when the reacting cell was irradiated under the AM 1.5 simulated sunlight. The hydrogen-producing rate measurement was carried out half-hourly by injecting 1 mL reacting gas into a Techcomp GC7900 gas chromatograph with a 15 Å molecular sieve packing column and a thermal conductivity detector. The circulating water system was applied to maintain the reacting cell at room temperature, avoiding the heat damage of the lasting light irradiation. A quantitative evaluation cycle lasted for 6 h. A fresh PBS solution was used for a new cycle.

## 4. Conclusions

In summary, a Pt_1_@TpPa-1/TiO_2_-NTAs photoelectrode was fabricated by the solvothermal growth of ultrathin TpPa-1 COF film on the amino-modified TiO_2_ NTAs/Ti foil and photo-deposition of Pt_1_. The well-defined TiO_2_ NTAs were easily obtained by a facile anodization method for 0.25 h and annealing at 450 °C for 2 h. Ultrathin TpPa-1 films could remarkably extend the absorption range and improve the photoconductance, interfacial resistance, and charge separation of TpPa-1/TiO_2_-NTAs heterostructures. With the single atom loaded, the Pt_1_@TpPa-1/TiO_2_-NTAs photoelectrode showed further enhanced PEC performance, lower overpotential (77 mV@10 mA cm^−2^), and reduced Tafel slope (61 mV dec^−1^). Under the optimized conditions, the hydrogen evolution rate of the Pt_1_@TpPa-1/TiO_2_-NTAs was 5 times that of the Pt_1_@TpPa-1 photoelectrode under AM 1.5 simulated sunlight irradiation and the bias voltage of 0 V. The excellent hydrogen evolution performance of Pt_1_@TpPa-1/TiO_2_-NTAs photoelectrode could be attributed to the well-matched band structures, effective charge separation, improved interfacial resistance, robust interfacial catalytic ability, and the hydrophilic surface of the photoelectrode. The convenient reusability and good stability of the Pt_1_@TpPa-1/TiO_2_-NTAs photoelectrode benefit its promising application in hydrogen evolution

## Figures and Tables

**Figure 1 molecules-28-00822-f001:**
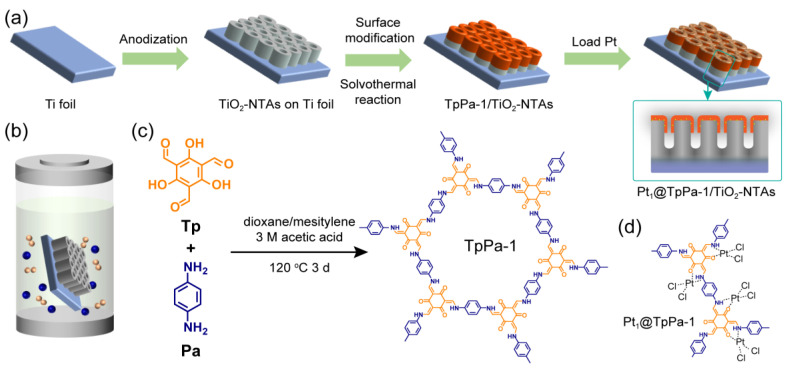
(**a**) Fabricating procedure of the Pt_1_@TpPa-1/TiO_2_-NTAs photoelectrode on the surface of Ti foil. (**b**) Schematic diagram of the solvothermal method for growing TpPa-1 film on amino-modified TiO_2_ NTAs. (**c**) Chemical structure of TpPa-1 COF condensing from monomers Tp and Pa. (**d**) Possible coordination structure of the photo-deposited Pt_1_ to TpPa-1.

**Figure 2 molecules-28-00822-f002:**
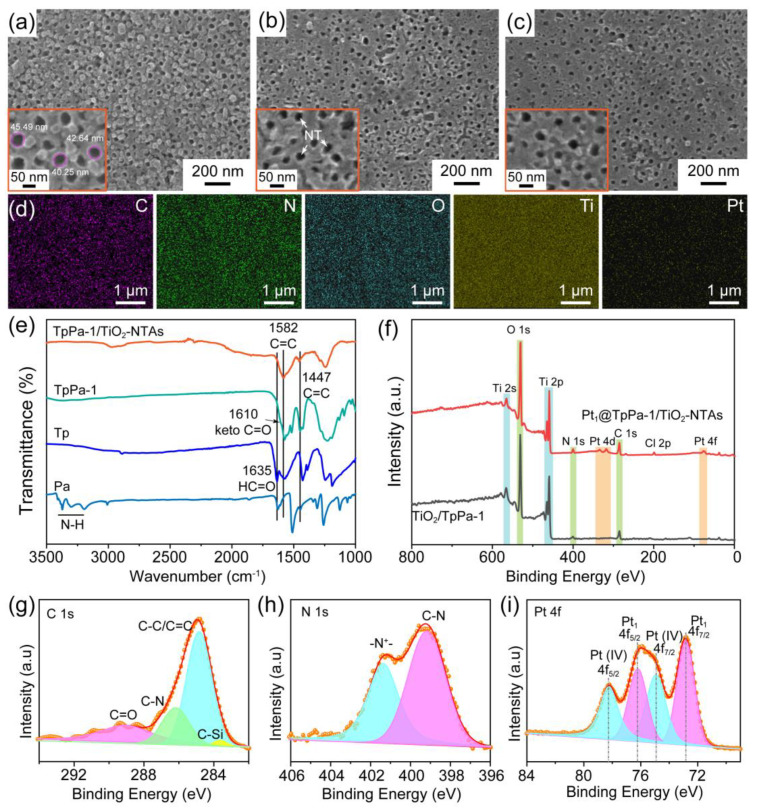
SEM images of (**a**) TiO_2_ NTAs on Ti foil, (**b**) TpPa-1/TiO_2_-NTAs, and (**c**) Pt_1_@TpPa-1/TiO_2_-NTAs. Insets: the amplified SEM images correspondingly. (**d**) Element mapping of Pt_1_@TpPa-1/TiO_2_-NTAs with SEM observation. (**e**) Comparison FTIR spectra of TpPa-1/TiO_2_-NTAs, TpPa-1 powders, and corresponding monomers (Tp and Pa). (**f**) XPS survey spectra of TpPa-1/TiO_2_-NTAs and Pt_1_@TpPa-1/TiO_2_-NTAs. High-resolution spectra of (**g**) C1s, (**h**) N1s, and (**i**) Pt 4f cores. The TpPa-1 COF films were obtained from 0.2 mM Tp and 0.3 mM Pa for the above characterization.

**Figure 3 molecules-28-00822-f003:**
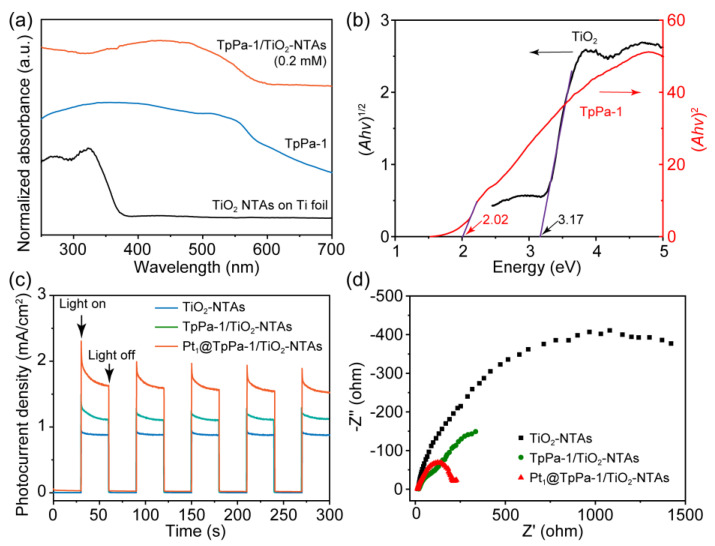
(**a**) UV–vis absorption spectra of TpPa-1/TiO_2_-NTAs, TpPa-1 powders, and TiO_2_ NTAs on Ti foils. (**b**) Tauc’s plots for TiO_2_ and TpPa-1 based on the absorption spectra. (**c**) Transient photocurrent response of TiO_2_ NTAs, TpPa-1/TiO_2_-NTAs, and Pt_1_@TpPa-1/TiO_2_-NTAs with or without light irradiation. (**d**) EIS Nyquist plots of TiO_2_, TpPa-1/TiO_2_-NTAs, and Pt_1_@TpPa-1/TiO_2_-NTAs under light irradiation. The TpPa-1 COF films were obtained from 0.2 mM Tp and 0.3 mM Pa for the above characterization.

**Figure 4 molecules-28-00822-f004:**
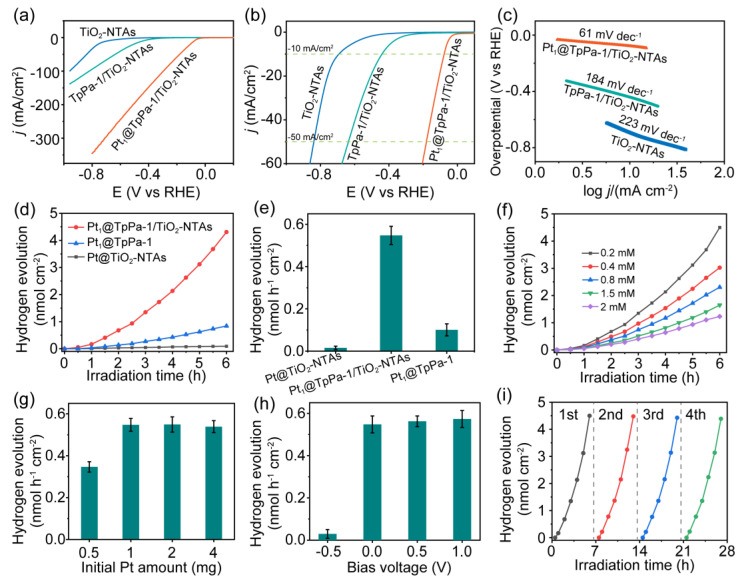
(**a**,**b**) Linear scan voltammetry curves and (**c**) Tafel plots of TiO_2_ NTAs, TpPa-1/TiO_2_-NTAs, and Pt_1_@TpPa-1/TiO_2_-NTAs under light irradiation. (**d**,**e**) Hydrogen evolution performance of Pt_1_@TpPa-1/TiO_2_-NTAs compared to Pt@TiO_2_ NTAs and Pt_1_@TpPa-1. Effect of (**f**) monomer concentration (based on Tp), (**g**) initial Pt amount, and (**h**) applied bias voltage (vs. SCE) on the hydrogen generation performance of Pt_1_@TpPa-1/TiO_2_-NTAs photoelectrode. (**i**) Cycle tests of hydrogen evolution.

**Figure 5 molecules-28-00822-f005:**
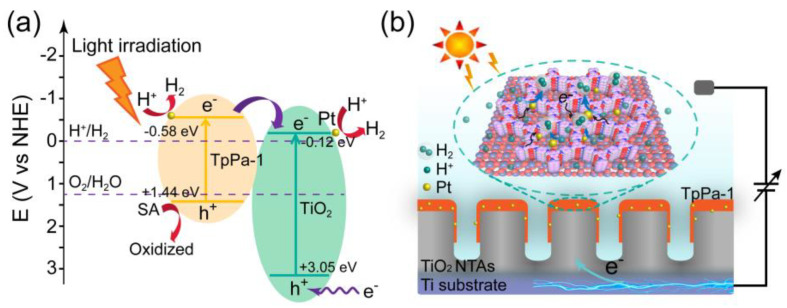
(**a**) Band diagram and possible charge transfer under light irradiation for the Pt_1_@TpPa-1/TiO_2_-NTAs photoelectrode. (**b**) Interfacial illustration for the photoelectrochemical hydrogen evolution by using the Pt_1_@TpPa-1/TiO_2_-NTAs photoelectrode.

## Data Availability

Most of the data presented in this study are available in the Appendix A. Additional data presented in this study are available on request from the corresponding authors.

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
