# Peer review of "A Heterostructure Photoelectrode Based on Two-Dimensional Covalent Organic Framework Film Decorated TiO2 Nanotube Arrays for Enhanced Photoelectrochemical Hydrogen Generation"

_molecules, 2023, doi:10.3390/molecules28020822_

Round 1

Reviewer 1 Report

In this work, authors developed sensitive and low-cost Pt1@TpPa-1/TiO2-NTAs photoelectrode by the solvothermal growth of ultrathin TpPa-1 COF fil for enhanced photoelectrochemical hydrogen geneneration. The authors report interesting results, with good suporting references, figures and tables. Although manuscript is being interesting and informative, I find that there are some issues that require addressing prior publication.

-        Author should elaborate does this photoelectrode have any interfering compounds in its ussuall aplications?

-        Elaborate on the photoelectrode lifetime, and heat resistance.

-        Elaborate in detail, and support with references which are the major advantages of Pt1@TpPa-1/TiO2-NTAs photoelectrode vs. other similar compounds.

-        At the end of introduction part, explain the main innovation of research, its impact and how it can help in the improvement of industrial applications?

For the above reasons I suggest to accept the manuscript after some minor revisions.

Reviewer 2 Report

 In this paper, the authors present heterostructures based on typical stable covalent organic framework (TpPa-1) films and TiO2 nanotube arrays (NTAs) for the photoelectrochemical hydrogen generation. They show that the Pt1@TpPa-1/TiO2-NTAs photoelectrode with single-atom Pt1 as co-catalyst has improved visible-light response, enhanced photoconductance, lower onset potential, and improved Tafel slope value for hydrogen evolution.

The study is useful and interesting. The paper is clear and well-organized. It reports a detailed experimental study that is carefully analyzed.

The manuscript can be suitable for publication after a revision.

 Here are my comments and suggestions:

 “Recently, two-dimensional covalent organic frameworks (2D COFs) have emerged as promising materials in hydrogen evolution due to their predictable structures, tunable π-conjugation, high crystallinity, and permanent pore channels.[8–11]” I suggest adding the following review paper on MOF: https://doi.org/10.3390/s22062238.

 “The tailored coordinating microenvironments of 2D COFs benefit the immobilization of single-atom noble metals to reduce the overpotential, promoting the hydrogen evolution procedure.[20,21]” Single atoms are cited here for the first time. The role of single-atom catalysts should be better emphasized. Perhaps a citation (see for instance https://doi.org/10.1149/1945-7111/ac62c3) could be enough.

 Figure 2: I wonder if the authors could add higher magnification SEM images for a), b) and c) to make the TiO2 nanotube arrays clearly distinguishable.

 “The typical discrete nanotubes are obtained with an average diameter of 40.55 ± 8.46 nm  after the anodization time of 0.25 h (Figure 2a).” How is the diameter of the nanotube measured? I suggest giving the experimental error with one or two significant digits.

 “The Pt1@TpPa-1/TiO2-NTAs photoelectrode is finally constructed by photo-depositing Pt species to TpPa-1/TiO2-NTAs photoelectrodes.” I wonder if the authors have tried any other single-atom other than Pt.
